# The Role of Social Media in Improving Patient Recruitment for Research Studies on Persistent Post-Infectious Olfactory Dysfunction

**DOI:** 10.3390/medicina58030348

**Published:** 2022-02-25

**Authors:** Alfonso Luca Pendolino, Annakan V. Navaratnam, Juman Nijim, Christine E. Kelly, Premjit S. Randhawa, Peter J. Andrews

**Affiliations:** 1Department of ENT, Royal National ENT & Eastman Dental Hospitals, London WC1E 6DG, UK; annakan.navaratnam@nhs.net (A.V.N.); prem.randhawa@nhs.net (P.S.R.); peterandrews@nhs.net (P.J.A.); 2Ear Institute, University College London, London WC1X 8EE, UK; 3Medical School, University College London, London WC1E 6DE, UK; juman.nijim@nhs.net; 4AbScent, Andover SP10 2PA, UK; chris@abscent.org

**Keywords:** smell, olfaction, olfactory dysfunction, social media, research

## Abstract

*Background and Objectives*: Since the COVID-19 pandemic, the number of cases of post-infectious olfactory dysfunction (PIOD) has substantially increased. Despite a good recovery rate, olfactory dysfunction (OD) becomes persistent in up to 15% of cases and further research is needed to find new treatment modalities for those patients who have not improved on currently available treatments. Social media has emerged as a potential avenue for patient recruitment, but its role in recruiting patients with smell dysfunction remains unexplored. We conducted a survey using the AbScent Facebook page to evaluate the feasibility of using this platform for future studies on smell dysfunction. *Materials and Methods*: Between 26 October and 4 November 2021, we conducted an online survey to evaluate propensity of patients with PIOD who would be willing to participate in research studies on smell dysfunction. *Results*: Sixty-five subjects were surveyed with a response rate of 90.7%. The median visual analogue scale (VAS) for sense of smell was 0 at infection and 2 at survey completion. The median length of OD was 1.6 years, and the main cause of OD was SARS-CoV-2 (57.6%). Parosmia was reported in 41 subjects (69.5%) whilst phantosmia in 22 (37.3%). The median length of olfactory training (OT) was 6 months but subjectively effective in 15 subjects (25.4%). Twenty-seven subjects (45.8%) tried other medications to improve olfaction, but only 6 participants (22.2%) reported an improvement. All subjects expressed their propensity to participate in future studies with most of them (38; 64.4%) willing to be enrolled either in medical and surgical studies or to be part of a randomised study design (11; 18.6%). *Conclusions*: Using the AbScent Facebook platform we successfully selected a population of subjects with persistent and severe OD that have failed to improve on available treatments and are willing to participate in further clinical trials.

## 1. Introduction

Post-infectious olfactory dysfunction (PIOD) is the second commonest cause (11%) of olfactory dysfunction (OD) after chronic rhinosinusitis and followed by trauma and idiopathic OD [1]. Since the coronavirus (COVID-19) pandemic, the number of cases of PIOD has substantially increased with smell loss being a common symptom (up to 65% of those infected) following SARS-CoV-2 infection. According to a recent systematic review, COVID-19-related OD recovers in most infected patients within the first month (recovery rate 94.6%). By 6 months the recovery rate is 85.7% [2] while at almost one year following infection this could be as high as 87.2% [3]. This means that in the remaining cases (up to 12.8%) OD becomes persistent.

European guidelines recommend olfactory training (OT) for a minimum of 3 months to maximise the chance of smell improvement [1]. It involves repeated daily exposure of a subject to a range of odourants [1]. Nonetheless, up to 29% of PIOD cases do not improve even after long-term OT (14 months) [4]. Other therapies for smell loss (i.e., corticosteroids, phosphodiesterase inhibitors, intranasal sodium citrate, intranasal vitamin A) have not been proven to be effective in PIOD and consequently are not recommended [5].

There is a desperate need to find new treatment modalities for patients with persistent PIOD who failed other available therapies. These subjects could be recruited into future research studies looking at smell restoration; however, an effective recruitment process is required.

Traditional methods for patient recruitment include offline media (non–Internet-based) platforms such as newspapers, radio or television advertisements, mailings, flyers and posters in hospitals and/or general practitioner offices [6]. Over the last few years, social media has emerged as a source for patient recruitment. Social media has the ability to target particular groups of patients and then uses customized messages to engage specific subjects. According to a recent scoping review, the use of social media does result in a higher recruitment and enrolment rate with a lower per-participant cost when compared to traditional recruitment methods [7]. Whilst the use of social media platforms has been documented as a recruitment tool into health, medical, and psychosocial research studies [8], its feasibility to recruit participants with smell disorders remains unexplored.

The challenge of using social networks to recruit patients with OD mainly relies on the subjectivity in which patients perceive their smell impairment in the absence of an objective (measurable) OD [9]. Moreover, OD may vary widely within affected subjects with some patients referring to only a qualitative olfactory impairment (e.g., parosmia– distorted sense of smell) in the absence of a reduced sense of smell [10]. Therefore, this high variability in the OD presentation must be taken into account when polling participants to avoid inclusion of potentially noneligible subjects.

To better investigate this, we conducted an online survey using the AbScent Facebook page with the aim of evaluating the feasibility of using this platform for patient recruitment in future research studies on the sense of smell.

## 2. Materials and Methods

This study was conducted in accordance with the 1996 Helsinki Declaration and approved by our research ethic committee (REC ref 14/SC/1180). The survey was carried out in collaboration with AbScent, a United Kingdom (UK) charity gathering people with smell disorders.

### 2.1. Setting of the Survey

Between 26 October and 4 November 2021, we conducted an online survey to evaluate propensity of patients with OD to take part in future research studies on the sense of smell (Appendix A). Patients were recruited using the AbScent Facebook platform which has more than 70,000 members who joined the group for information and support about their smell loss. We included patients who were living in the United Kingdom having a PIOD (reduced or absent sense of smell for more than 12 months) with or without parosmia and having tried smell training for at least 3 months. Patients who reported parosmia as the only symptom were excluded. Subjects satisfying the above criteria were entered into the study with their consent and provided their details (name, surname and email) which were sent to the research team. A link to an online survey was then sent to the selected respondents along with a brief description of the survey aims. Before being able to start the questionnaire, subjects were also required to sign the online consent form and to satisfy the entry criteria. The survey was performed via Survey Monkey (San Mateo, CA, USA) and the questionnaire was validated locally by both ENT clinicians and patient advocates to ensure clarity and to exclude ambiguity.

### 2.2. Population and Data Collection

The recipients of this survey were members of the AbScent Facebook group with persistent PIOD which did not improve following a minimum of 3 months of OT. Data were collected anonymously mainly on OD characteristics, different treatments tried to improve olfaction and propensity to take part in research studies. Basic demographic data were also collected for all the participants to investigate characteristics of the population polled. Quantitative variables were summarized using median and interquartile range (P25–P75) while qualitative variables were described with frequency and percentage. Missing values (i.e., people who did not answer the question) were not considered in the calculation of percentages (valid percent). However, unanswered items have been reported in the tables.

## 3. Results

### 3.1. Response Rate and Population Characteristics

Eighty-six subjects fulfilled the initial inclusion criteria and received the email survey invitation of which 78 accessed the link (response rate 90.7%). After further analysis, we excluded 13 participants who did not pass the entry criteria, leading to a final population of 65 subjects. The total population was composed of 18 men and 47 women (male to female ratio approximately of 1:3), ranging from 13 to 76 years, with a median age of 55 years (Table 1).

### 3.2. Characteristics of Olfactory Dysfunction

The median length of OD was 1.6 years, and the main cause for smell loss was SARS-CoV-2 (57.6%). Parosmia was reported by 41 subjects (69.5%), whilst phantosmia was reported by 22 participants (37.3%). The median Visual Analogue Scale (VAS) [11] score for sense of smell was 0 at the time of infection and 2 when they completed the survey. Smell was reported to improve at the moment of the survey in only 9 respondents (15.3%), whilst for the majority (35; 59.3%) smell had stopped improving (Table 1).

### 3.3. Treatments Tried for Olfactory Dysfunction

The median length of olfactory training (OT) was 6 months. Fifteen subjects (25.4%) found their sense of smell improved during OT; however, the majority (25; 42.4%) reported no benefit. The remaining (19; 32.2%) were not sure it was effective. Twenty-seven subjects (45.8%) tried other medications to improve olfaction (Table 1). Of all the treatments mentioned by the subjects, topical nasal steroid spray was the most used treatment (12; 44.4%) followed by oral steroids (11; 40.7%) and topical nasal steroid drops (7; 25.9%). Only 6 participants (22.2%) felt the treatment they tried improved the sense of smell, and in all the cases, the treatment deemed to be effective was the combination of oral steroids and topical steroid drops (Table 2).

### 3.4. Propensity to Take Part in Research Study

All subjects expressed their inclination to take part in future research studies with the majority (38; 64.4%) willing to be enrolled either in medical and surgical studies or to be randomised (11; 18.6%) (Table 3).

## 4. Discussion

To the best of our knowledge, this is the first study to evaluate the use of social media as a tool to enroll patients into potential research studies on smell restoration. Social media represents a powerful source to increase awareness of on-going research studies, further promote patient understanding and improve recruitment. The ability to specifically connect with a target group of patients through collaboration with specific charities such as AbScent can increase the probability of successful recruitment. We have demonstrated this through our survey response rate of 90.7%. These platforms can capture a higher proportion of subjects of different age groups and from minority or marginalized groups [12,13], thus improving patient representation. Even though we did not collect information on ethnicity, our observed wide range in age distribution (13 to 76 years) confirms the wide-reaching effects of social media. The higher representation of female participants in our study group is not a consequence of survey bias but reflects the higher proportion of female subjects commonly affected by PIOD [1,14].

To avoid recruitment of potentially non-enrollable subjects, an appropriate screening process is recommended. A clear description of the eligibility criteria should be outlined in the study information page before starting any online recruitment process. As a further precaution, we decided to add these inclusion criteria to the beginning of the online survey to highlight essential entry requirements. We observed that 13 subjects (13/78; 16.7%) were automatically logged out because they did not meet at least one of the entry criteria, despite having read the information page and considering themselves as eligible. This highlights the importance of a two-level selection process.

Social media represents an important tool to collect data on the sense of smell. Very recently, Koyama and colleagues [15] analyzed the posts and comments on a social media membership group and confirmed its usefulness in understanding and investigating COVID-19 symptoms, especially focusing on anosmia and ageusia. In this regard, when enrolling subjects with an OD, obtaining an idea of the level of their smell impairment is of paramount importance to avoid the inclusion of possible false positives. However, this could be challenging considering the subjectivity of olfaction. Moreover, people with OD may experience either quantitative or qualitative disorders, the latter including parosmia (distorted perception of an odour) and phantosmia (perception of an odour in the absence of a stimulus) [1]. The situation becomes even more complex if we consider that during the smell recovery phase, qualitative OD (especially parosmia) may become the main reported symptom [16,17,18] and potentially mask a possible smell improvement. Patient reported outcome measures (PROMs) represent a good way to measure quantitative OD when this cannot be assessed by using psychophysical tests (e.g., Sniffin’ Sticks).

Even though it is well-known that self-reported OD poorly correlates with olfactory tests [19] and PROMs could be unreliable when used to assess smell recovery in the long-term [9], they remain of value given their good discriminative ability [20]. Considering it would be almost impossible to assess olfaction using Sniffin’ Stick evaluation (average time 30–40 min per test), an accurate screening tool becomes crucial and, in this regard, PROMs play an important role. We suggest the use of the VAS for sense of smell as a tool to easily and quickly self-rate olfaction. In our population, the VAS score for sense of smell at the moment of the survey was very low (median value of 2 after a median period of 1.6 years). When compared to the VAS at the time of infection (median value of 0), the level of improvement was low, thus demonstrating a persistent poor sense of smell.

It is also important to understand whether subjects have failed recommended treatments and/or tried other therapies before asking them to participate in experimental studies. So far, OT remains the only treatment recommended for both COVID-19 and non-COVID-19-related OD [1,5], but its long-term efficacy remains unclear [4]. Our results show that after an average length of 6 months of OT, only 25.4% of subjects noticed a clear improvement. The role of other treatments in PIOD remains unproven. In our population 45.8% of the subjects (*n* = 27) tried other treatments ranging from supplements to oral steroid, but only six of them (22.2%) found these to be helpful. Interestingly, these six subjects reported the combination of oral plus topical (drops) steroids to be effective in improving their sense of smell. This reflects recent findings showing a potential role of steroids in improving the sense of smell following PIOD [21,22].

According to currently available studies, a high proportion of subjects do not recover their sense of smell long-term following PIOD [23]. In the specific case of COVID-19-related OD, which represented the highest reported OD cause (57.6%) in our polled population, the long-term recovery rate is not available yet even though up to 12.8% of subjects may complain of persistent PIOD one year after their infection [3]. Based on the extension of the present pandemic, a significant number of patients with persistent PIOD is expected, and considering the exiguity of available treatments, further research is warranted. Engaging patients since the initial phases of the study makes them feel that their contributions, feedback and participation are valued and acknowledged. This may finally translate to a better recruitment process. Our results show that all participants in the survey were interested in taking part in future research studies on the sense of smell with most of them having no preferences on the type of study (medical or surgical) or to be randomised as part of the study design. The high propensity of subjects with OD to be involved in research studies could be partially due to the high impact OD has on quality of life. Smell loss is highly debilitating, and up to 76% of these subjects suffer from depression [24]. The inability to recognise spoiled food, hazardous odours such as gas leaks, smoke or undetected volatile chemicals also poses an additional safety risk [25].

## 5. Conclusions

Social media may improve patients’ engagement into research, and our results suggest that this is a useful tool to identify appropriate subjects for future studies on smell restoration. Evaluation of the sense of smell is key in the early stages of the enrolment process, and we recommend the use of the VAS scale to perform a quick screening of suitable candidates before psychophysical testing. Using the AbScent Facebook platform, we successfully selected a population of subjects with PIOD who had failed maximal therapy and were willing to take part in future clinical trials.

## Figures and Tables

**Table 1 medicina-58-00348-t001:** Characteristics of the population.

	Subjects (*n* = 65)
Age, median (P25–P75), year	55 (46–63)
Sex, No (%)	
Female	47 (72.3%)
Male	18 (27.7%)
Length of OD, median (P25–P75), year	1.6 (1.3–1.7)
OD cause, No (%)	
SARS-CoV-2 (tested positive)	34 (57.6%)
Not sure (never tested)	13 (22.0%)
Other viral infection ^1^	12 (20.4%)
Missing	6
Qualitative OD symptoms, No (%)	
Parosmia	41 (69.5%)
Phantosmia	22 (37.3%)
Both	20 (33.9%)
Missing	6
sVAS at infection, median (P25–P75)	0.0 (0.0–0.0)
sVAS at survey, median (P25–P75)	2.0 (1.0–3.0)
Smell reported to improve, No (%)	
Yes	9 (15.3%)
No	35 (59.3%)
Not sure	15 (25.4%)
Missing	6
Length of OT, median (P25–P75), month	6.0 (4.0–11.5)
Reported improvement on OT, No (%)	
Yes	15 (25.4%)
No	25 (42.4%)
Not sure	19 (32.2%)
Missing	6
Tried medications to improve olfaction, No (%)	
Yes	27 (45.8%)
No	32 (54.2%)
Missing	6

^1^ Tested negative for COVID-19 or OD before COVID-19 pandemic. Valid percentage, not including missing values: OD: olfactory dysfunction; OT: olfactory training; sVAS: visual analogue scale for sense of smell (0: Anosmia; 10: Normal sense of smell).

**Table 2 medicina-58-00348-t002:** Treatments tried excluding OT.

	Subjects (*n* = 27)
Medications tried to improve olfaction, No (%)	
Supplements ^1^	6 (22.2%)
Topical steroid (spray)	12 (44.4%)
Topical steroid (drops)	7 (25.9%)
Oral steroid	11 (40.7%)
Oral steroid + topical steroid (drops)	6 (22.2%)
Others (Vitamin A or nasal rinse)	3 (11.1%)
More than 2 medications	12 (44.4%)
Benefit reported on medication, No (%)	
Yes	6 (22.2%)
No	19 (70.4%)
Not sure	2 (7.4%)
Medications felt to improve olfaction, No (%)	*n* = 6
Oral steroid + topical steroid (drops)	6 (100%)

^1^ Including multivitamins, zinc, omega-3, lipoic acid.

**Table 3 medicina-58-00348-t003:** Inclination to take part in research studies.

	Subjects (*n* = 59) ^1^
Willing to take part, No (%)	
Yes	59 (100 %)
No	0 (0.0%)
Type of study, No (%)	
Medical	20 (33.9%)
Surgical	1 (1.7%)
Both	38 (64.4%)
Randomisation, No (%)	
Yes	48 (81.4%)
No	11 (18.6%)

^1^ 6 subjects missing.

## Data Availability

The data presented in this study are available on request from the senior author (P.J.A.). The data are not publicly available due to privacy reasons.

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
