# Peer review of "The Role of Social Media in Improving Patient Recruitment for Research Studies on Persistent Post-Infectious Olfactory Dysfunction"

_medicina, 2022, doi:10.3390/medicina58030348_

Round 1

Reviewer 1 Report

Overall, the authors present a survey done on those with olfactory deficits, in which the participants were recruited through Facebook. This study is timely and of interest since it is dealing with one of the long-lasting effects of COVID-19. There are a few recommendations that would strengthen this paper.

Abstract:

  • Line 14: The word 'the' should appear between Since and COVID-19.
  • The abbreviation VAS (line 24) should be defined. 
  • The abbreviation OT (line 26) should be defined.

Introduction:

  • Highly recommended: A short description of what olfactory training involves. This may help explain why it hasn't worked.
  • Throughout the Introduction and elsewhere, the number of the citation in-text should be BEFORE the period, not after. Line 46, for example, the [1] should be before the period and after the word improvement. This needs to be done throughout the text. 
  • Line 55: I believe the authors wanted to say general practitioner offices; the word surgeries has an odd placement here.

Materials & Methods:

  • Was the VAS done as part of the Survey Monkey survey? This was unclear. 
  • Is it possible to publish the entire survey the participants completed in this paper, or at least as Supplementary data?

Results:

  • Line 116: the parentheses is not closed after the work smell). 
  • There should be a citation for the VAS or it should be stated that this is a standardized smell test. It should be made available to the reader to see.
  • When referring to Tables, they should come before the period of the preceding sentence, not after the period. This should be done for all Table references in the Results.

Discussion:

  • For some reason the word Authors appears on line 144 right at the beginning of the Discussion. It should be deleted. 

Conclusions:

  • Line 221, there should be a comma between the word platform, and we successfully.

Author Response

In response to Reviewer #1:

  • Overall, the authors present a survey done on those with olfactory deficits, in which the participants were recruited through Facebook. This study is timely and of interest since it is dealing with one of the long-lasting effects of COVID-19. There are a few recommendations that would strengthen this paper.

- We thank the reviewer for his/her appreciation of our paper.

  • Abstract:
  • Line 14: The word 'the' should appear between Since and COVID-19.
  • The abbreviation VAS (line 24) should be defined.
  • The abbreviation OT (line 26) should be defined.

We thank the reviewer for his/her comments. The above acronyms have been defined in the abstract and the article ‘the’ has been added in line 14. (Lines 14, 24, 27)

  • Introduction:
  • Highly recommended: A short description of what olfactory training involves. This may help explain why it hasn't worked.
  • Throughout the Introduction and elsewhere, the number of the citation in-text should be BEFORE the period, not after. Line 46, for example, the [1] should be before the period and after the word improvement. This needs to be done throughout the text.
  • Line 55: I believe the authors wanted to say general practitioner offices; the word ‘surgeries’ has an odd placement here.

We thank the reviewer for his/her comments.

  • A short description of what olfactory training is has been added in the text (Lines 48-49).
  • The numbers referencing the citations in-text have now been put before the end of the corresponding sentence, as suggested.
  • The term ‘surgeries’ has been changed to ‘offices’ has suggested (Line 59).

  • Materials & Methods:

Was the VAS done as part of the Survey Monkey survey? This was unclear. Is it possible to publish the entire survey the participants completed in this paper, or at least as Supplementary data?

- We thank the reviewer for his/her comments. Yes, the VAS was part of the survey. As requested, the survey has now been uploaded as ‘supplementary data’ which should also make the point about VAS clearer. (Supplementary data.pdf) (Line 86)

  • Results:
  • Line 116: the parentheses is not closed after the work smell).
  • There should be a citation for the VAS or it should be stated that this is a standardized smell test. It should be made available to the reader to see.
  • When referring to Tables, they should come before the period of the preceding sentence, not after the period. This should be done for all Table references in the Results.

- We thank the reviewer for his/her comments.

  • Line 116 (now 123): correction has been made
  • A citation has been added to reference VAS scale when first mentioned in the text. (Line 128)
  • The word ‘table’ when referencing a table in the text has now been moved within the corresponding sentence. This correction has been made throughout the text.

  • Discussion:

For some reason the word Authors appears on line 144 right at the beginning of the Discussion. It should be deleted.

- Thanks for spotting this mistake. It has now been deleted. (Line 168)

  • Conclusions:

Line 221, there should be a comma between the word platform, and we successfully.

- We thank the reviewer for his/her comments. A comma has now been added (Line 247)

Reviewer 2 Report

The article “The Role of Social Media in Improving Patient Recruitment for Research Studies on Persistent Post-infectious Olfactory Dys-Function” by A.L.Pendolino, A.V.Navaratnam, J.Nijim, C.E.Kelly, P.S.Randhawa and P.J.Andrews aims to evaluate the feasibility of using a social media platform for patient recruitment for research studies on sense of smell.

Social media groups are a valuable tool for accessing large patient groups to understand or investigate the symptoms of mild to moderate COVID-19 patients, especially focusing on anosmia and ageusia. There are only a few papers dealing with the reports of patients obtained from social media. However, social media is undoubtedly a very valuable and important tool for obtaining outpatient data, even more so in an era marked by infectious pandemics, where many patients with severe conditions are piling up in hospitals and patients with mild or moderate symptoms are overlooked.

In this context, we are faced in this manuscript with a current and innovative approach of high interest to readers and researchers. The present study is based on a simple design, but very clear in its methodology, with a good typification of the results and the conclusions drawn. The authors convincingly demonstrate that the Facebook platform can be used to successfully select a population of patients willing to take part in future clinical trials.

More specifically, the study has the additional interest of providing results on the validity of olfactory training in a large sample of subjects and on the potential role of steroids in improving sense of smell following PIOD. These results are undoubtedly of high clinical interest.

In this sense, it would have been interesting to include a survey on cognitive disorders, as anosmia is commonly associated with brain clouding in the long-COVID. The evaluation of the coexistence of headache, migraine, dizziness, vertigo, depression, anxiety, insomnia, etc. could enrich the results of the study.

Apart from the minor issues raised at the end, my main objection to the paper is that it omits a similar study on olfactory disease based on the use of social networks. Specifically, this is the study by Koyama S, Ueha R, Kondo K. Loss of Smell and Taste in Patients With Suspected COVID-19: Analyses of Patients' Reports on Social Media. J Med Internet Res 2021;23(4):e26459. doi: 10.2196/26459

Although this is a different approach, various aspects such as gender and age, composition of the sample, treatments performed, degree of persistence of olfactory dysfunction, etc. are amenable to comparison in the discussion with the Koama et al. article. In any case, it is relevant to address this study in both the introduction and the discussion of the manuscript.

Minor issues:

Abstract:

Line 24: The abbreviation Vas should be defined as Visual Analogue Scale

Line 25: The abbreviation OT should be defined as Olfactory training

Introduction

Line 47: Other therapies should be very briefly described.

Line 55: A reference to traditional patient recruitment techniques is pertinent. For instance, Thoma, A., Farrokhyar, F., McKnight, L., & Bhandari, M. (2010). Practical tips for surgical research: how to optimize patient recruitment. Canadian journal of surgery. Journal Canadien de Chirurgie, 53(3), 205–210.

Line 144: The word authors should be deleted

Table 1:

Regarding the population characteristics: Would it not be of interest to take into account the geographical origin of patients?

Also, given the sociological character of the study, it would be interesting to have asked patients the reasons to join the membership group.

------------------------------------------------------------------------------------

Author Response

In response to Reviewer #2:

 [1]   The article “The Role of Social Media in Improving Patient Recruitment for Research Studies on Persistent Post-infectious Olfactory DysFunction” by A.L.Pendolino, A.V.Navaratnam, J.Nijim, C.E.Kelly, P.S.Randhawa and P.J.Andrews aims to evaluate the feasibility of using a social media platform for patient recruitment for research studies on sense of smell.Social media groups are a valuable tool for accessing large patient groups to understand or investigate the symptoms of mild to moderate COVID-19 patients, especially focusing on anosmia and ageusia. There are only a few papers dealing with the reports of patients obtained from social media. However, social media is undoubtedly a very valuable and important tool for obtaining outpatient data, even more so in an era marked by infectious pandemics, where many patients with severe conditions are piling up in hospitals and patients with mild or moderate symptoms are overlooked.In this context, we are faced in this manuscript with a current and innovative approach of high interest to readers and researchers. The present study is based on a simple design, but very clear in its methodology, with a good typification of the results and the conclusions drawn. The authors convincingly demonstrate that the Facebook platform can be used to successfully select a population of patients willing to take part in future clinical trials.More specifically, the study has the additional interest of providing results on the validity of olfactory training in a large sample of subjects and on the potential role of steroids in improving sense of smell following PIOD. These results are undoubtedly of high clinical interest.In this sense, it would have been interesting to include a survey on cognitive disorders, as anosmia is commonly associated with brain clouding in the long-COVID. The evaluation of the coexistence of headache, migraine, dizziness, vertigo, depression, anxiety, insomnia, etc. could enrich the results of the study. 

- We thank the reviewer for his/her appreciation of our paper and we totally agree with their comments. We normally assess cognitive impairment in patients with olfactory dysfunction coming to our smell clinic for a face-to-face appointment. OD highly impacts on quality of life and cognitive function. It increases patient’s morbidity as well as mortality. However, cognitive evaluation was not included in the survey as it was beyond the aim of our paper which mainly focused on selecting a population of subjects with persistent and severe OD that have failed to improve on available treatments and willing to participate in further clinical trials. However, I agree that this interesting aspect should be studied in patients with OD when assessing their sense of smell and it should be further evaluated in future research.

 [2]   Apart from the minor issues raised at the end, my main objection to the paper is that it omits a similar study on olfactory disease based on the use of social networks. Specifically, this is the study by Koyama S, Ueha R, Kondo K. Loss of Smell and Taste in Patients With Suspected COVID-19: Analyses of Patients' Reports on Social Media. J Med Internet Res 2021;23(4):e26459. doi: 10.2196/26459Although this is a different approach, various aspects such as gender and age, composition of the sample, treatments performed, degree of persistence of olfactory dysfunction, etc. are amenable to comparison in the discussion with the Koyama et al. article. In any case, it is relevant to address this study in both the introduction and the discussion of the manuscript. 

- We thank the reviewer for his/her comments. The paper from Koyama et al aims “to analyze the posts and comments on this social media membership group to establish the benefits and limits of using social media to understand COVID-19 and to investigate the symptoms of mild to moderate COVID-19 patients, especially focusing on anosmia and ageusia”. Conversely, we wanted to evaluate feasibility of using a Facebook platform to recruit patients with COVID-19 related olfactory dysfunction in future research studies on sense of smell. Even though the populations considered in the article are similar (i.e. participants of a social media group), the aims and conclusions are different. However, I agree it is appropriate to discuss and reference this paper in the discussion. (Line 189-192)

[3]   Abstract:

  • Line 24: The abbreviation Vas should be defined as Visual Analogue Scale
  • Line 25: The abbreviation OT should be defined as Olfactory training. 

- We thank the reviewer for his/her comments. The above acronyms have been defined in the abstract. (Lines 24, 27)

[4]   Introduction 

  • Line 47: Other therapies should be very briefly described.
  • Line 55: A reference to traditional patient recruitment techniques is pertinent. For instance, Thoma, A., Farrokhyar, F., McKnight, L., & Bhandari, M. (2010). Practical tips for surgical research: how to optimize patient recruitment. Canadian journal of surgery. Journal Canadien de Chirurgie, 53(3), 205–210.
  • Line 144: The word authors should be deleted 

- We thank the reviewer for his/her comments.

  • Other therapies have now been mentioned in the text (Lines 50-51)
  • The suggested paper by Thoma et al. has now been referenced in the text (Line 59) 
  • The word ‘authors’ has been deleted. Thanks for spotting this mistake. (Now line 168)

[5]   Table 1:

  • Regarding the population characteristics: Would it not be of interest to take into account the geographical origin of patients?
  • Also, given the sociological character of the study, it would be interesting to have asked patients the reasons to join the membership group. 

- We thank the reviewer for his/her comments.

  • As part of the inclusion criteria to take part in this survey, all patients had to be UK citizens and living in the UK. Further information about their provenience, were not requested. The reason for this inclusion criterion was that we wanted to select a population of patients with persistent COVID-19-related olfactory dysfunction potentially enrollable in research studies in the UK. This was reported in the materials and methods section (Lines 88-90)
  • Regarding your point about “reasons to join the membership group” I discussed that with the founder of the AbScent Facebook platform, who is also one of the co-authors. Because of the nature of the Facebook platform, there is no facility for asking people why they joined. On the AbScent Facebook page they normally ask two questions to new members: “Have you lost your sense of smell?” and “Do you agree to admin rules?”. Members all join for info and support about smell loss. This information has been added in the text. (Lines 87-88)  

This manuscript is a resubmission of an earlier submission. The following is a list of the peer review reports and author responses from that submission.